# Effect of Ellagic Acid on Fermentation Quality and Bacterial Community of Stylo Silage

**Xuan Zou [1], Dandan Chen [1], Hongjian Lv [1], Qing Zhang [1,\*] and Peng Zheng [2,\*]**

1. Guangdong Province Research Center of Woody Forage Engineering Technology, College of Forestry and Landscape Architecture, South China Agricultural University, Guangzhou 510642, China; zxzouxuan@stu.scau.edu.cn (X.Z.); 1744301540@stu.scau.edu.cn (D.C.); hongjian@stu.scau.edu.cn (H.L.)
2. College of Horticulture, South China Agricultural University, Guangzhou 510642, China
* Correspondence: zqing@scau.edu.cn (Q.Z.); zhengp@scau.edu.cn (P.Z.)

**Abstract:** This study was conducted to investigate the effect of ellagic acid on the bacterial community and fermentability of stylo silage. Three treatments of stylo silage were used: control (CK) and treated with 1% or 2% ellagic acid (EA1 and EA2) on a fresh matter basis. All silage was stored at ambient temperature and opened on days 3, 7, 14, and 30. Fermentation characteristics, protein fraction, and bacteria community of all periods of silage were analyzed. Results showed that dry matter and crude protein content were increased, and pH value, number of coliform bacteria, contents of acetic acid, and ammonium nitrogen were decreased with the addition of ellagic acid. The antioxidant activity of 1% and 2% ellagic acid treated silages was significantly higher than the control. Meanwhile, the relative abundance of *Klebsiella* and *Clostridium* was decreased with the addition of ellagic acid, and the abundance of *Lactobacillus*, *Weissella*, and *Enterococcus* was increased with prolonged days of ensiling. Adding ellagic acid to stylo silage could improve the fermentation quality and preservation of protein, and reduce the abundance of harmful bacteria.

**Keywords:** bacterial community; stylo; ellagic acid; fermentation quality; silage

## 1. Introduction

Ensiling is a common method for the preservation of fresh forage. It can both reduce environmental risk, and also provide an extended period of availability of nutritious and palatable feed for livestock [1]. It is generally understood that high-quality silage can easily be made when dry matter content reaches 30–35%, water-soluble carbohydrate content reaches 60–80 g/kg, and lactic acid bacteria numbers exceed 5.00 log10·cfu/g FM (log10·cfu/g Fresh Matter) in raw materials [2–4]. Stylo (*Stylosanthes guianensis* Sw.), known as pencilflower, a common flowering legume that is native to South America, grows mainly in tropical and subtropical regions. In the subtropics, it is considered an important feed source for ruminants, with high yields, high nutrient levels, and wide adaptability [5]. Although stylo occupies an important position in the market, its production is seasonal; the long dry seasons and low regeneration rate can easily lead to a shortage of livestock feed [6], therefore, storing it as silage would be highly beneficial. Silage quality is affected by many factors, legumes like stylo are difficult to ensile directly without additives because of their high buffering capacity, low water-soluble carbohydrate content, and high dry matter content [7,8]. Moreover, undesirable microorganisms such as *Clostridia* and *Enterobacter* are always present in legume silage during ensiling, which leads to butyric acid accumulation and proteolysis [9]. Therefore, the use of additives should assist in making high-quality stylo silage.

Ellagic acid (EA, $C_{14}H_6O_8$), is a plant polyphenol. Free EA, EA derivatives, and bound forms as ellagitannins occur naturally in many economic plant species, particularly fruits like berries and nuts [10]. According to a previous report, there are high levels of ellagitannins and allagic acid in strawberry, cranberry, blueberry, and blackberry [10].

Previous studies have demonstrated the great potentials of EA. Epidemiological evidence suggests that the intake of EA-rich foods may be protective against certain chronic diseases. Furthermore, the antiproliferative and apoptosis-inducing activities of EA have been demonstrated to inhibit cancer cell growth [10]. Moreover, many studies report that pomegranate extracts, which contain abundant EA could reduce cancer, inflammation, and harmful bacteria [11–13]. Ellagitannin and punicalagin have similar characteristics and they all correlate closely with EA, which could inhibit the growth of pathogenic organisms [14,15]. This potential antimicrobial ability of EA might be helpful in improving the microbial community of silage. Pomegranate fruit husks or peels are inexpensive and abundant sources of hydrolyzable tannins called ellagitannins (ETs). Alternately, EA could be obtained from the degradation of punicalagin (a type of ellagitannin). "Seeram et al." reported that using analytical HPLC and tandem LC-ES/MS to evaluate total pomegranate tannins (TPT) showed that it contains the major fruit husk ETs, punicalagin, and EA, and unquantified amounts of punicalin and EA-glycosides (hexoside, rhamnoside, and pentoside). This method can be used for the large-scale production of TPT, and could be practical for industrial applications to provide a low-cost means of obtaining EA. Furthermore, Gil et al. and Kotsampasi et al. report that using pomegranate by-product silage to partially replace alfalfa hay in the diet of lamb could improve the antioxidative potential, and the nutritional and functional qualities of meat [16–18]. Kotsampasi [19] later reported that adding pomegranate pulp silage to the diets of cows could improve milk fatty acid profiles and animal antioxidant status. The positive effect of adding pomegranate products might be owing to the high content of EA in pomegranate. Many studies focus on how additives affect the fermentation quality of silage, but less on their antioxidant capacity. It is known that excess free radicals cause oxidative stress, which is related to many diseases in the human body, such as cancer, autoimmune disorders, aging, cataracts, rheumatoid arthritis, and cardiovascular and neurodegenerative diseases [20].

From the above information, and because of its chemical and biological characteristics, it is believed that the addition of EA would inhibit the activity of undesirable microorganisms and have similar positive effects during the ensiling process. However, to date, little information is available about the effects of EA application on silage. We hypothesized that EA would positively affect silage quality. In the present study, stylo was ensiled with 1% and 2% EA, and fermentation quality and microbial community were analyzed at different ensiling periods (days 3, 7, 14, and 30).

## 2. Materials and Methods

### 2.1. Raw Material and Silage Preparation

Stylo was planted in an experimental field at South China Agricultural University (Guangzhou, China) and harvested manually in October, 2019. The laboratory replicates for mini-silo research were according to the method of the Editorial note by Robinson et al. [21]. The raw materials were chopped to about 2 cm in length by a crop chopper and then randomly allocated to 39 subsamples (about 120 g for each subsample). Three subsamples were stored at −20 °C for further analysis. The remaining 36 subsamples were assigned to one of the following treatments: (1) the control group with no additives (CK); (2) 1% ellagic acid (EA1); (3) 2% ellagic acid (EA2). The additives were applied on the basis of fresh weight, and the application rate was determined according to He et al. [22]. The materials were packed into plastic silo bags, which were vacuumed and sealed with a vacuum sealer. Each treatment included 12 mini-silos that were stored at ambient temperature. Three mini-silos of each treatment were opened at days 3, 7, 14, and 30 and the fermentation characteristics, protein fraction, and bacteria community were analyzed. The experiment had a completely randomized design with three treatments, four ensiling periods, and three replicates.

## 2.2. Microbial and Chemical Composition

According to Wang et al. [3], 20 g samples (raw materials and silage) were immediately blended with 180 mL sterile normal saline solution and serially diluted from $10^{-1}$ to $10^{-6}$. The numbers of lactic acid bacteria (LAB), coliform bacteria, yeasts, and molds were incubated and counted using Man Rogosa Sharpe (MRS) agar, Violet Red Bile agar, and Rose Bengal agar, respectively. The colony counts indicated the numbers of viable microorganisms in cfu/g FM. Furthermore, contents of organic acids and ammonium nitrogen ($NH_3$-N), and pH value were determined by separate 20 g samples that were blended with 180 mL distilled water for 18 h at 4 °C. The remaining samples were oven-dried at 65 °C for 48 h for dry matter (DM) determination and ground to determine protein fractions (crude protein, true protein, and non-protein nitrogen). Crude protein (CP) was analyzed using a Kjeldahl nitrogen analyser (Kjeltec 2300 Auto-Analyser, FOSS Analytical AB, Hoganas, Sweden) according to the methods of the Association of Official Analytical Chemists [22].

## 2.3. Bacteria Community Analysis

Total DNA in silage was extracted with the E.Z.N.A. stool DNA Kit (Omega Biotek, Norcross, GA, US) following the manufacturer's protocol. PCRs were conducted in a 50 μL mixture, including 5 μL of 2.5 mM dNTPs, 5 μL of 10 × KOD buffer, 1.5 μL of each primer (5 μM), 1 μL of KOD polymerase, and 100 ng of template DNA. According to Wang et al. [23] and Wang et al. [24], the V3–V4 regions of 16S rDNA were amplified, sequenced, and analyzed.

After purification and quantification, the PCR products were sequenced using an Illumina platform (Guangzhou Gene Denovo Co. Ltd., Guangzhou, China). The raw sequences were selected according to Wang et al. [23]. Paired-end clean reads were merged as raw tags using FLASH (v.1.2.11) with a minimum overlap of 10 bp and mismatch error rates of 2%. Noisy sequences filtering and data processing were performed using QIIME (v.1.9.1). Clean tags were searched against the reference database (http://drive5.com/uchime/uchime_download.html) to perform reference-based chimera checking using the UCHIME algorithm (http://www.drive5.com/usearch/manual/uchime_algo.html) in March 2021. Chimeric sequences were removed and the effective tags with 0.97 identities were clustered into operational taxonomic units (OTU) using the UPARSE pipeline. The analysis of taxonomy assignment of representative sequences was performed using Ribosome Database Project (RDP) classifier (v.2.2). Finally, functional genes of the bacterial communities were predicted using Tax4Fun [25]. The sequencing data were submitted to the National Center for Biotechnology Information Sequence Read Archive database under the BioProject accession number PRJNA718453.

## 2.4. Antioxidant Activity

Approximately 0.2 g sample powder was extracted with 10 mL of methanol in a 15 mL plastic tube. Following vigorous shaking, the tube remained in a shaker incubator (200 r/min, room temperature) for 24 h. Following centrifugation (3000 r/min), the extract was collected from the mixture [24].

The 2,2-diphenyl-1-picrylhydrazyl (DPPH) assay was conducted according to the methods of He et al. (2019b) with some modifications. A 0.5 mL extract (properly diluted) or standard solution of Trolox (10–80 mg/L; $R^2$ = 0.996) was added to 4 mL of freshly prepared 0.1 mM DPPH (methanol solution). The mixture was well agitated and then stored in the dark for 30 min. The absorbance was read at 517 nm. The radical scavenging activity was expressed as mg Trolox equivalents (TE)/g dry matter.

The 2,2-azinobis-3-ethylbenzothiazoline-6-sulfonic acid diammonium salt radical cation (ABTS) assay was used according to Abdennacer et al. [26] with slight modifications. Similar volumes of ABTS solution (7 mM) and potassium persulfate solution (2.45 mM) were mixed to make the stock solution and it was stored in the dark for 16 h at room temperature. A mixture of 0.2 mL extract or standard solution of Trolox (10–250 mg/L;

$R^2 = 0.9919$) and 4 mL of ABTS diluted solution ($\times 20$) was kept in the dark for 6 min. The absorbance of the solution was read at 734 nm. The radical scavenging activity was expressed as mg Trolox equivalents (TE)/g dry matter.

The ferric-reducing antioxidant power (FRAP) assay was used according to Li et al. [27]. The FRAP working solution was freshly made by mixing together 10 mM TPTZ and 20 mM FeCl3 in 0.25 M HOAc-NaOAc buffer (pH 3.6) at a ratio of 1:1:10. A mixture of 0.5-mL of extract or standard solution of Trolox (10–90 mg/L, $R^2 = 0.9943$) and 4 mL of the FRAP solution was stored at 37 °C for 30 min. The absorbance of the solution was read at 593 nm. The reducing power was expressed as mg Trolox equivalent (TE)/g dry matter.

The total flavonoid content was determined by the aluminum chloride colorimetric assay according to He et al. [24] with some modifications. A 0.5 mL extract or standard solution of rutin (100–1000 mg/L; $R^2 = 0.9952$) was mixed with 0.15 mL 5% (wt/vol) $NaNO_2$ and kept for 6 min. It was then mixed with 0.15 mL 10% (wt/vol) $AlCl_3$ and kept for 6 min. Subsequently, 2.2 mL 4% (wt/vol) NaOH was added and kept for a further 6 min. Finally, 2.2 mL of distilled water was added to make a total mixture of 5 mL. The absorbance of the mixture was read at 510 nm. Total flavonoid content was expressed as mg rutin equivalent (RE)/g dry matter.

### 2.5. Statistical Analysis

The effects of adding EA, ensiling period, and their interaction were analyzed using two-way analysis of variance. All statistical procedures were performed using SPSS 19.0 for Windows (SPSS, Chicago, IL, USA). Values of $p < 0.05$ and $p < 0.01$ were considered statistically significant and highly significant, respectively.

## 3. Results and Discussion

### 3.1. Characteristics of Fresh Stylo

Chemical composition and microbial population are shown in Table 1. The dry matter of fresh stylo was 34.3%, which was relatively high compared with the findings of He et al. [22], and it reached the ideal dry matter content (30–35%) for good silage [2]. The CP, true protein (TP), and non-protein nitrogen (NPN) content of fresh stylo were 13.3%, 7.67%, and 5.60% DM, respectively. The CP content (13.3% DM) was comparable with the data of Wang et al. [23], of which 58.0% was TP. In general, a higher proportion of TP indicates better nutritional value of the protein given that NPN is less efficiently utilized in ruminants relative to true protein [22]. The NDF and ADF contents were 60.2% and 47.0% DM, respectively, which means a little high fraction of fiber, and they were higher than the data of He et al. [22]. These differences in the stylo might be because forage quality is influenced by factors such as climate and fertilization [28]. The numbers of lactic acid bacteria (LAB), coliform bacteria, yeast, and molds were 4.93, 5.68, 3.83, and 3.71 log10·cfu/g FM, respectively. The number of LAB almost reached the threshold (>5.00 log10·cfu/g FM) for well-preserved silage [4]. However, the relatively high numbers of undesirable bacteria might cause difficulty in making high-quality silage. Thus, using additives to improve the silage quality is necessary.

**Table 1.** Characteristics of fresh stylo ($n = 3$, $\pm$SD).

| Items | Stylo |
|---|---|
| Dry matter (DM, %) | $34.3 \pm 0.67$ |
| Water-soluble carbohydrates (WSC, %DM) | $4.94 \pm 0.67$ |
| Crud protein (CP, %DM) | $13.3 \pm 0.99$ |
| True protein (% TN) | $58.0 \pm 3.75$ |
| Non-protein nitrogen (% TN) | $42.00 \pm 3.75$ |
| Neutral detergent fiber (NDF, %DM) | $60.2 \pm 0.54$ |
| Acid detergent fiber (ADF, %DM) | $47.0 \pm 0.09$ |
| Lactic acid bacteria (LAB, log10·cfu/g FM) | $4.93 \pm 0.19$ |
| Yeasts (log10·cfu/g FM) | $3.83 \pm 0.35$ |
| Coliform bacteria (log10·cfu/g FM) | $5.68 \pm 0.18$ |
| Molds (log10·cfu/g FM) | $3.71 \pm 0.30$ [1] |

[1] cfu, colony forming units; FM, fresh matter

### 3.2. Fermentation Quality of Stylo Silage

Dry matter recovery, pH value, organic acid content, and microbial population of stylo silage without/with the addition of EA are listed in Table 2. The dry matter recovery of stylo silage with EA was significantly increased, and this might improve stylo silage quality; as reported by McDonald et al. [4] as the chances of *Clostridia* fermentation are minimized by ensiling forages with more than 30% DM. It is well known that pH declines during ensiling as a result of the generation of organic acids and this is affected greatly by the buffering capacity of raw materials. In the present study, both pH values and acetic acid content decreased, but lactic acid content remained unchanged; though pH values showed a decrease following the addition of EA and the prolongation of ensiling days, they were still higher than the common threshold (pH 4.2) of well-fermented silage [29], which might not be beneficial for aerobic stability and long-time preservation. These low levels might be ascribed to their low organic acid content, which was likely caused by the low WSC content in stylo [22]. In effect, the high pH values would have created the high buffering capacity, which would have led to low organic acid content and counteraction of pH decline [22]. Adding EA can decrease the number of coliform bacteria; furthermore, coliform bacteria decrease with prolongation of ensiling days as shown by the dynamic changes in our previous study [30]. The presence of abundant lactic acid bacteria might be owing to the prebiotic effects of EA [31], while the number of coliform bacteria might be explained by the effect of the high pH. Moreover, Hayrapetyan et al. [32] reported that the antimicrobial activity of pomegranate peel extract was less pronounced at higher temperatures (7 and 12 °C), which might indicate that EA did not effectively inhibit coliform bacteria at relatively high temperatures. Acetic acid is always converted from lactic acid because of the presence of *Enterobacter*; therefore, the decrease in acetic acid might be identical with the decrease of coliform bacteria.

**Table 2.** Fermentation quality of stylo silage treated with or without ellagic acid.

| Item | Treatments | Ensiling Days | | | | Means | SEM | p-Value | | |
|---|---|---|---|---|---|---|---|---|---|---|
| | | 3 | 7 | 14 | 30 | | | D | T | D*T |
| Dry matter (DM, %) | CK | 34.0 B | 34.9 C | 34.5 B | 35.1 B | 34.6 C | | | | |
| | EA1 | 35.7 A | 35.9 B | 35.5 AB | 35.7 AB | 35.7 B | 0.09 | * | ** | NS |
| | EA2 | 35.5 A | 36.7 A | 36.4 A | 36.7 A | 36.7 A | | | | |
| Dry matter recovery (%) | CK | 98.7 B | 101.1 C | 99.8 C | 100.3 B | 100.0 C | | | | |
| | EA1 | 103.8 Ba | 103.8 Ba | 102.4 Bb | 101.8 ABb | 103.0 B | 0.22 | NS | ** | NS |
| | EA2 | 104.3 A | 106.1 A | 106.4 A | 105.1 A | 105.5 A | | | | |
| pH | CK | 5.65 b | 5.94 Aa | 5.48 Ac | 5.29 Ad | 5.62 A | | | | |
| | EA1 | 5.53 b | 5.89 Ba | 5.44 Ab | 5.18 Bc | 5.47 B | 0.01 | ** | ** | NS |
| | EA2 | 5.53 b | 5.80 Ca | 5.29 Bc | 5.07 Cd | 5.41 C | | | | |
| Lactic acid (LA, g/kg) | CK | 1.62 | 1.65 | 1.67 | 1.70 | 1.66 | | | | |
| | EA1 | 1.62 | 1.63 | 1.70 | 1.72 | 1.67 | 0.000 | ** | NS | NS |
| | EA2 | 1.62 | 1.65 | 1.68 | 1.74 | 1.67 | | | | |
| Acetic acid (AA, g/kg) | CK | 0.27 | 0.62 A | 0.96 | 1.77 | 0.97 A | | | | |
| | EA1 | 0.28 | 0.43 B | 0.90 | 1.68 | 0.82 B | 0.002 | ** | ** | NS |
| | EA2 | 0.23 | 0.37 B | 0.71 | 1.57 | 0.72 C | | | | |
| Propionic acid (PA, g/kg) | CK | ND | ND | ND | ND | - | - | - | - | - |
| | EA1 | ND | ND | ND | ND | - | - | - | - | - |
| | EA2 | ND | ND | ND | ND | - | - | - | - | - |
| Butyric acid (BA, g/kg) | CK | ND | ND | ND | ND | - | - | - | - | - |
| | EA1 | ND | ND | ND | ND | - | - | - | - | - |
| | EA2 | ND | ND | ND | ND | - | - | - | - | - |
| Lactic acid bacteria (LAB, log10·cfu/g FM) | CK | 7.93 a | 7.71 b | 7.27 c | 7.20 c | 7.52 | | | | |
| | EA1 | 7.59 ab | 7.94 b | 7.35 b | 7.30 b | 7.55 | 0.03 | ** | NS | * |
| | EA2 | 7.86 a | 7.64 ab | 7.49 b | 7.12 c | 7.53 | | | | |
| Coliform bacteria (log10·cfu/g FM) | CK | 7.10 Ab | 7.48 a | 6.56 c | 4.72 d | 6.62 A | | | | |
| | EA1 | 6.59 Bb | 7.73 a | 6.65 b | 3.74 c | 6.18 B | 0.05 | ** | ** | ** |
| | EA2 | 6.29 Ca | 7.38 a | 6.63 b | 3.15 d | 6.11 B | | | | |
| Yeasts (log10·cfu/g FM) | CK | 2.62 | <2.00 | <2.00 | 2.95 | 2.80 | | | | |
| | EA1 | <2.00 | 3.01 | <2.00 | 2.45 | 2.73 | 0.21 | NS | NS | NS [2] |
| | EA2 | 3.52 | <2.00 | 2.54 | 2.00 | 2.69 | | | | |

[2] CK was the control group with no additive, EA1 and EA2 were treated with 1% and 2% ellagic acid on a fresh matter basis; values in the same row (or same column) with different superscripts in lowercase letters (or capital letters) differ significantly from each other at $p < 0.05$; "ND" means no detected; "NS" means "no significance"; "*" and "**" mean significant at $p < 0.05$ and 0.01, respectively; DM, dry matter, SEM, standard error of the mean.

### 3.3. Protein Content of Stylo Silage

Proteolysis may be an issue in silage production. As shown in Table 3, EA had a positive influence on the preservation of protein. For ruminants, non-protein nitrogen is less efficient in nitrogen retention than true protein, thus extensive proteolysis would lead to inferior nutritional value of the silage and more nitrogen emission during animal production [22]. The accumulation of ammonium nitrogen in silage is typically caused by the synthetic effects of plant protease activity and microbial activity [33]. In the present study, though the $NH_3$-N content was very low, adding EA could have contributed to decreasing the $NH_3$-N content. EA is a hydrolysate of ellagitannin and naturally occurring or exogenous tannins have been shown to reduce proteolysis of alfalfa silage [34,35]. With the prolongation of ensiling days, $NH_3$-N levels were increased; this also took place in the studies of He et al. [36] and Li et al. [37]. In summary, adding EA could help to reduce levels of harmful bacteria and proteolysis in stylo silage.

**Table 3.** Protein content of stylo silage treated with or without ellagic acid.

| Item | Treatment | Ensiling Days | | | | Means | SEM | p-Value | | |
|---|---|---|---|---|---|---|---|---|---|---|
| | | 3 | 7 | 14 | 30 | | | D | T | D*T |
| Crude protein (% DM) | CK | 11.1 [Ba] | 9.56 [Bb] | 10.0 [b] | 8.94 [c] | 9.91 [B] | 0.08 | ** | * | * |
| | EA1 | 12.1 [Aa] | 9.82 [Bb] | 10.8 [b] | 9.34 [b] | 10.4 [A] | | | | |
| | EA2 | 10.8 [Ba] | 10.8 [Aa] | 10.7 [a] | 9.08 [b] | 10.3 [A] | | | | |
| True protein (% TN) | CK | 51.9 [A] | 52.4 | 50.9 | 52.0 | 51.8 | 0.58 | NS | NS | NS |
| | EA1 | 45.2 [B] | 56.3 | 49.4 | 51.3 | 51.05 | | | | |
| | EA2 | 54.3 [Aa] | 48.8 [b] | 47.3 [b] | 55.4 [a] | 51.5 | | | | |
| Non-protein nitrogen (% TN) | CK | 48.1 [B] | 47.6 | 49.1 | 48.0 | 48.2 | 0.58 | NS | NS | ** |
| | EA1 | 54.7 [A] | 43.7 | 50.6 | 48.7 | 48.9 | | | | |
| | EA2 | 45.7 [Bb] | 51.2 [a] | 52.7 [a] | 44.6 [b] | 48.5 | | | | |
| Ammonium nitrogen (% TN) | CK | 2.61 [d] | 5.07 [Ac] | 6.31 [b] | 9.05 [a] | 5.76 [A] | 0.14 | ** | ** | NS |
| | EA1 | 1.97 [d] | 3.59 [Bc] | 5.62 [b] | 8.6 [a] | 4.95 [B] | | | | |
| | EA2 | 2.19 [d] | 2.99 [Bc] | 4.55 [b] | 7.54 [a] | 4.51 [B] | | | | |

CK was the control group with no additive, EA1 and EA2 contained 1% and 2% ellagic acid; values in the same row (or same column) with different superscripts in lowercase letters (or capital letters) differ significantly from each other at $p < 0.05$; "NS" means "no significance"; "*" and "**" mean significant at $p < 0.05$ and 0.01, respectively; TN, total nitrogen, DM, dry matter, SEM, standard error of the mean.

### 3.4. Antioxidant Activity

The antioxidant activity of stylo silage treated with EA is shown in Figure 1. As the figure shows, when stylo silage was treated with EA, the Trolox equivalent antioxidant capacity (TEAC) value and FRAP value were significantly higher than CK. Throughout the whole ensiling period, the DPPH scavenging activity of EA1 and EA2 were approximately 5- and 10-fold greater than the CK. The ABTS free radical scavenging activity of EA1 and EA2 were approximately 2- and 3-fold greater than the CK. The FRAP values of EA1 and EA2 were approximately 4- and 10-fold greater than the CK. The total flavonoid content was increased ($p < 0.05$) when treated with EA, but there was no significant effect at D7 ($p > 0.05$). The antioxidant activity was relatively low in raw stylo material. Cohen-Zinder et al. [38] reported that the ensiling process could reduce the polyphenol concentration of crops, which might cause a reduction in antioxidant activity. In the present study, the ensiling process created a small decrease in the antioxidant activity of the control group. The antioxidant activity of the EA treated groups was remarkably higher than CK. The addition of EA increased the antioxidant activity because of its characteristics. Studies report that EA has anti-cancer and anti-inflammatory properties, which might be a result of its powerful antioxidant activity [11,39]. Pomegranate byproduct silage is rich in EA, and has been shown to improve milk and goat meat profiles and antioxidative status [18,19]. Moreover, EA is found in many types of plants and fruits [18,19], few studies have reported the antioxidant activity of silage with additives, actually antioxidant activity is related to the quality of agricultural product, animal diseases, and aging problems.

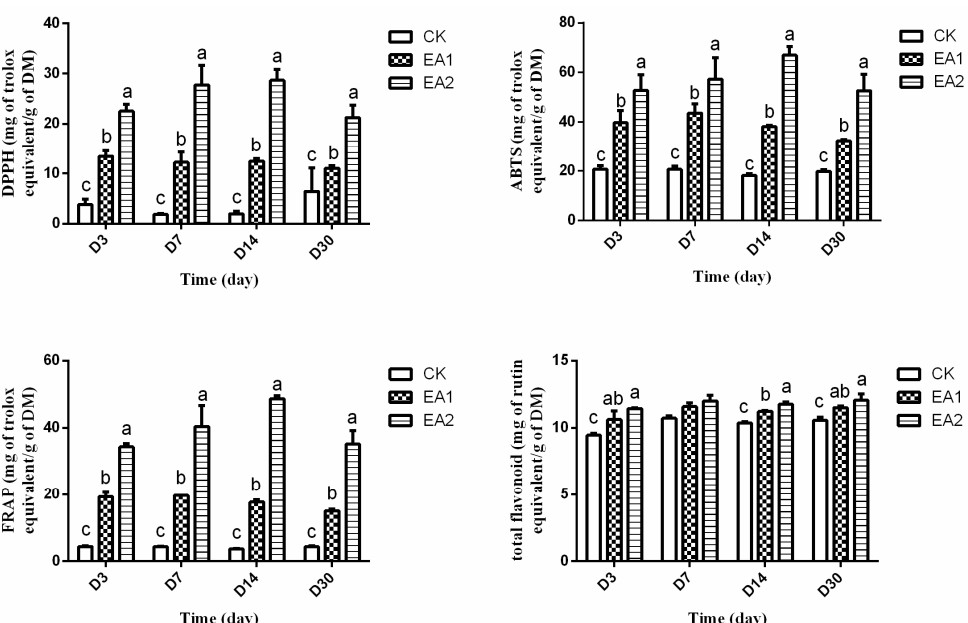

**Figure 1.** 2,2-diphenyl-1-picrylhydrazyl (DPPH), ferric-reducing antioxidant power (FRAP), and ferric-reducing antioxidant power (FRAP); CK was the control group with no additive, EA1 and EA2 included 1% and 2% ellagic acid, respectively; D3, D7, D14, and D30 were silage at days 3, 7, 14, and 30, respectively. The DPPH and ABTS scavenging activity, FRAP reducing power, and total flavonoid of silages treated with or without ellagic acid were expressed as Trolox equivalent (mg of TE/g DM) and rutin equivalent (mg of TE/g DM). Different superscripts in lowercase letters differ significantly from each other at $p < 0.05$.

### 3.5. Bacteria Community

The variance in bacterial community structure of stylo silage was demonstrated by principal coordinate analysis (Figure 2). The variances of bacterial community among the three treatments became clear with prolonged ensiling days, which might partially explain the enhancement of fermentation in EA treatments. In the whole ensiling period, the proportion of *Proteobacteria* was more than 65%. In the early stage, *Cyanobacteria* was second highest, but it was replaced by *Firmicutes* in the later stage. A similar phenomenon was shown in the studies of Liu et al. [40], Lv et al. [30], and Yang et al. [41]. Principal component 1 (PC1) and 2 (PC2) explained 46.52% and 23.7% of the total variance, respectively. The distinctions of bacterial communities were clear after 30 days of ensiling, and the separation became more obvious with prolonged ensiling days. The bacterial community structure of stylo silage at phylum and genus level are shown in Figures 3 and 4. As shown in Figure 3, the components of bacterial community were similar at the phylum level, with all groups mainly containing *Proteobacteria*, *Cyanobacteria*, *Firmicutes*, and *Actinobacteria*. Among these, *Proteobacteria* was the most abundant bacteria in every group (66.67–84.49%), and it decreased when treated with EA at D3 and D30. *Cyanobacteria* (4.26–18.17%) was the second most abundant, it showed a decrease in the overall level with prolongation of ensiling days. *Firmicutes* (4.95–16.71%) increased with prolongation of ensiling days, while it was decreased by adding EA at D3, D14, and D30. It was noted that *Cyanobacteria* was increased in EA treated stylo silage at D30. *Cronobacter spp.* are emerging opportunistic human pathogens mostly found in plant sources [42]. The high abundance of *Cronobacter* has been shown in other studies of stylo silage [23]. Though *Cronobacter* was only decreased by adding EA at D14, the decrease with prolonged ensiling days was still a good sign. Perhaps more measures should be taken to restrict this genus. *Methylobacterium* are facultative methylotrophic bacteria that are commonly found in plants [43]. Figure 4 lists *Cronobacter* (2.01–5.86%), *Methylobacterium* (2.12–4.05%), and *Enterococcus* (0.97–4.18%) as the three most dominant bacteria in the present study. From D3 to D30, the proportions of *Cronobacter* and *Methylobacterium* first increased and then decreased to the lowest level. The proportion of

*Enterococcus* was increased with prolongation of ensiling days. At D3, D7, and D14, adding EA decreased the abundance of *Klebsiella* while increasing the abundance of *Kosakonia*. The abundance of *Enterobacter* changed irregularly within a range of 1.40–3.17% during the ensiling period, which reached a peak at D14. The abundance of *Rhizobia* was in the range of 0.81–2.07%, and was increased in the groups treated with 2% EA. *Lachnoclostridium* abundance increased from 0.22–0.38% at D3 to 0.68–1.90% at D30. *Clostridium* abundance decreased from 0.55–2.18% at D3 to 0.10–0.43% at D30. *Pantoea* was increased in EA-treated groups including D3-EA2, D7-EA1, D7-EA2, D30-EA1, and D30-EA2.

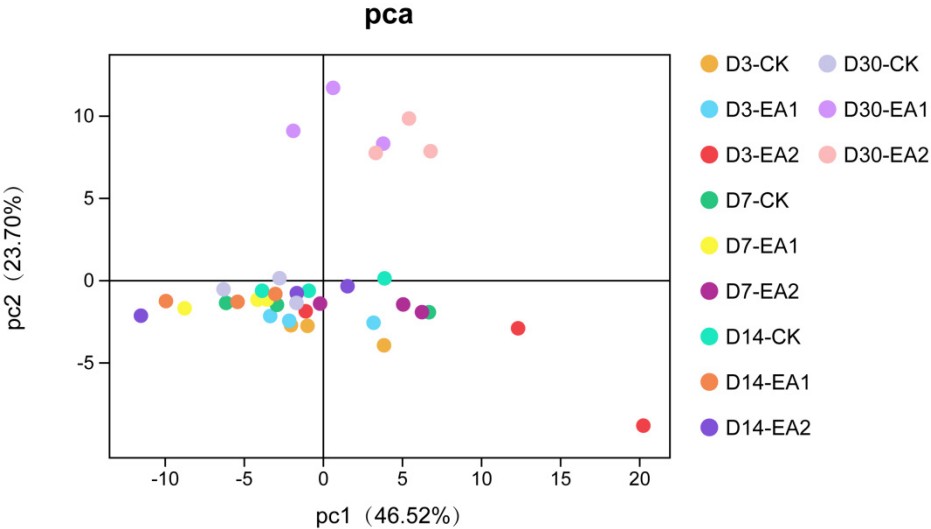

**Figure 2.** Principle component analysis of stylo silage (CK, the control; EA1, 1% ellagic acid; EA2, 2% ellagic acid; D3, D7, D14, and D30, after 3, 7, 14, and 30 days of ensiling, respectively).

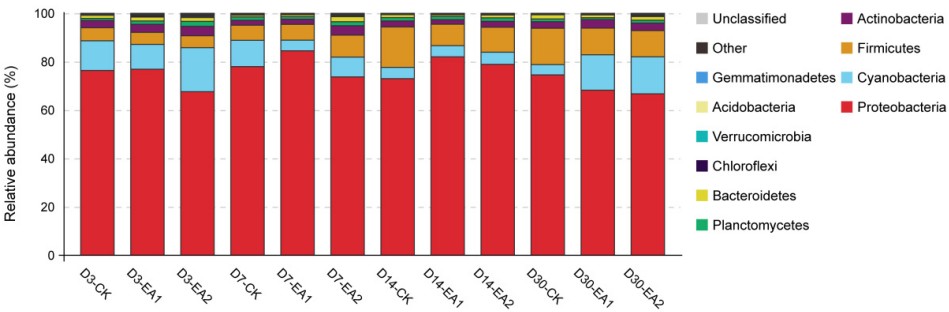

**Figure 3.** Bacterial community and relative abundance by phylum for stylo silage (CK, the control; EA1, 1% ellagic acid; EA2, 2% ellagic acid; D3, D7, D14, and D30, after 3, 7, 14, and 30 days of ensiling, respectively).

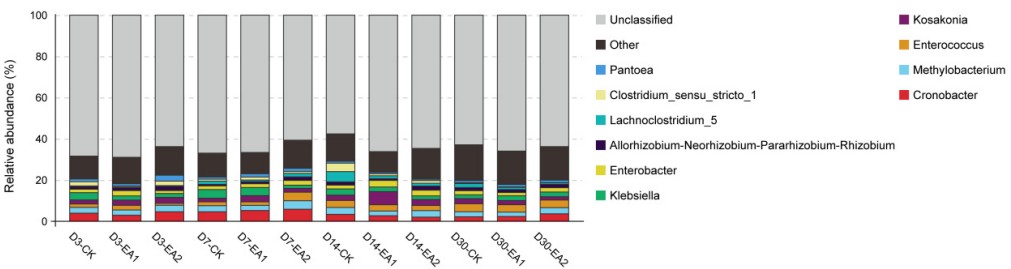

**Figure 4.** Bacterial community and relative abundance by genus for stylo silage (CK, the control; EA1, 1% ellagic acid; EA2, 2% ellagic acid; D3, D7, D14, and D30, after 3, 7, 14, and 30 days of ensiling, respectively).

Previous studies showed different abundances of *Methylobacterium* in silage, approximately 11% in He et al. [22] and about 1% in Ogunade et al. [44]. Ogunade et al. [44] reported that *Methylobacterium* is positively correlated with silage pH, which did tally with the present study. From D3 to D30, pH decreased from 5.53–5.65 to 5.07–5.29, while the abundance of *Methylobacterium* decreased from 2.57–3.15% to 2.12–2.93%. It may be that the relatively high pH value gave it a suitable condition to thrive and increase; therefore, decreasing the pH value or using EA with a proper dose may be useful in controlling the abundance of this genus. *Enterococcus* spp. are lactic acid bacteria that are useful in improving fermentation quality and are usually present in silage [41,45]. Desirable lactic acid bacteria dominated at D30, its increase might be the cause of pH decrease. According to Figure 5, the relative abundance of *Lactobacillus*, *Weissella*, and *Enterococcus* were increased at D14 and D30. The relative abundance of *Lachnoclostridium* and *Clostridium* showed a decrease at D14 and D30 when EA was added.

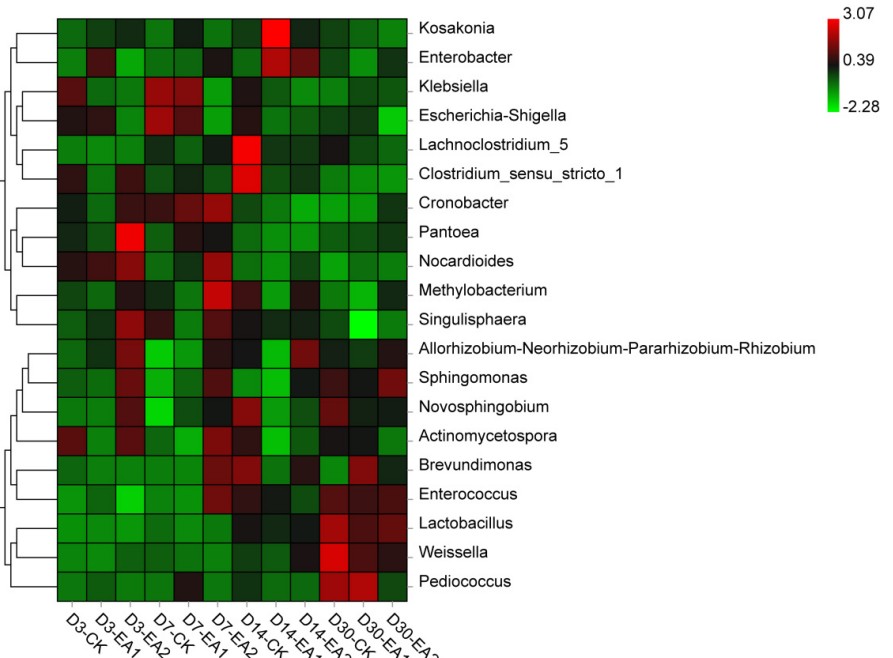

**Figure 5.** Heatmap of prominent bacterial genera (20 most abundant genera) for stylo silage (CK, the control; EA1, 1% ellagic acid; EA2, 2% ellagic acid; D3, D7, D14, and D30, after 3, 7, 14, and 30 days of ensiling, respectively).

*Kosakonia* is newly classified from the genus *Enterobacter*, and may be helpful in reducing the production of NH$_3$-N [41,45]. Corresponding results were shown in the present study; the NH$_3$-N content decreased when the abundance of *Kosakonia* was increased by adding EA at D3, D7, and D14. *Klebsiella* is a pathogen commonly detected in silage, it can also cause mastitis, reduced milk yields, and lead to subsequent culling on dairy farms [30]. Its decrease might be owing to the antibacterial ability of EA. The reasons for reverse relative abundances of *Kosakonia* and *Klebsiella* on D30 were not clear; however, desirable results were shown in the EA treated groups at the ensiling period except at D30. *Enterobacter* is undesirable as it may cause NH$_3$-N increase and compete with LAB for nutrients. In the present study, its abundance did not show a clear relationship with the addition of EA or the ensiling period.

*Lachnoclostridium* are rod-shaped *Clostridia* that grow in the conditions of 20–63 °C and neutral to alkaline pH; their fermentation product is acetate [46]. The main effect of *Lachnoclostridium* on silage is not clear, they may function like *Clostridia* based on their phyletic classification. *Clostridium* usually cause protein loss and an accumulation of acetic acid. However, in the present study, acetic acid contents were very low in each group and *Clostridium* abundance had already decreased to a very low level at D30. Additionally,

*Clostridium* was almost replaced by *Lachnoclostridium* during the ensiling period, it appears that they were negatively correlated in the silages. A positive sign was that the EA treated groups had lower relative abundance of *Lachnoclostridium* and *Clostridium* at D14 and D30.

The role of *Pantoea* in silage fermentation is still unclear. Ogunade et al. [44] found negative correlations between Pantoea and $NH_3$-N content, and inferred the genus could decrease $NH_3$-N during ensiling. On the contrary, Li et al. thought the role of *Pantoea* in silage was similar to that of *Enterobacter*, concluding that it was also undesirable because it would compete for substrates with LAB. However, the abundance of *Pantoea* might have been too low to have an effect in the present study. Members of the genus *Rhizobia* (*Allorhizobium*, *Neorhizobium*, *Pararhizobium*, *Rhizobium*) were found in the present study, they are usually found in soils planted with legumes, and they assist legumes with nitrogen fixation. However, studies have rarely reported the functions of *Rhizobia* in silage fermentation; more studies should be conducted to investigate their roles. In the present study, more attention should be paid to the unclassified genus, which could better explain the effects of adding EA on stylo silage. The increase of *Lactobacillus*, *Weissella,* and *Enterococcus* and the decrease of *Klebsiella* indicate that the fermentation quality was improving in the later ensiling period.

## 4. Conclusions

Ellagic acid had positive effects on fermentation quality and the bacterial community of stylo silage. The present study showed that crude protein content was increased, and pH value, number of coliform bacteria, and contents of acetic acid and ammonium nitrogen were decreased following the addition of EA. Meanwhile, the relative abundance of *Klebsiella* and *Clostridium* were decreased with the addition of EA, and the abundance of *Enterococcus* was increased with prolonged ensiling days. After ensiling for 14 days, the increase of *Lactobacillus*, *Weissella,* and *Enterococcus* might indicate that useful microbials were taking a dominant position. These results suggest that EA could be a novel additive to improve fermentation quality and protein preservation of stylo silage.

**Author Contributions:** Data curation, X.Z., H.L. and P.Z.; Investigation, D.C.; Project administration, X.Z.; Resources, Q.Z.; Writing—original draft, X.Z.; Writing—review & editing, H.L. and Q.Z. All authors have read and agreed to the published version of the manuscript.

**Funding:** This work was supported by Guangdong Natural Science Foundation (2020A1515011253).

**Institutional Review Board Statement:** Not applicable.

**Informed Consent Statement:** Not applicable.

**Data Availability Statement:** Not applicable.

**Conflicts of Interest:** The authors declare no conflict of interest.

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
