# Peer review of "Effect of Ellagic Acid on Fermentation Quality and Bacterial Community of Stylo Silage"

_fermentation, doi:10.3390/fermentation7040256_

Round 1
Reviewer 1 Report
Zou et al., studied the feasibility of ellagic acid as an additive for stylo silage preparation. The manuscript contains good data and can be a good reference for the community. However, the manuscript should be rewritten to be a concise one. It is difficult to follow. Introduction is not focused so the hypothesis is not clear. Discussion should not be dissected into sections. Moreover, discussion in the manuscript is more like a rephrase of results and contains no discussion. The topic is interesting and has potential practical usage. The manuscript is highly recommended to be revised into a short and concise one.
Author Response
Thank you very much for the professional opinions of the review experts. We are sorry for the unprofessional writing of the article. Now we have revised the manuscript, and we look forward to your reply.

Reviewer 2 Report
Please see the attached document.

Author Response
Response to Reviewer 2 Comments
Point 1: Page 1, lines 11-14 – I didn't understand the rationality of using numbers in
parentheses.
Response 1: Thank you for your suggestion. We have deleted these numbers and highlighted in yellow in line 12-14.
Point 2: Page 1, Abstract - Please don't use so many abbreviations in the abstract. They make the text difficult to read.
Response 2: Thank you very much for your professional opinion. We have modified it and highlighted it in yellow in abstract .
Point 3: Page 1, lines 17-19 - Please check the designation “vs” in this context.
Response 3: Thank you very much for your professional opinion. We have deleted the data that is not clear so that can understand it better and highlighted in yellow in line 16-18.
Point 4: Page 1, line 32 - Give a transcript of the abbreviation FM.
Response 4: Thank you very much for your reminding. We have completed transcript of the abbreviation FM and highlighted in yellow in line 32.
Point 5: Page 1, line 39 - Give the systematic name of the acid and its chemical formula.
Response 5: Thank you very much for your professional opinion. We've added the chemical formula for the acid and highlighted in yellow in line 45.
Point 6: Page 1, line 42 - Provide a reference.
Response 6: Thank you very much for your reminding. References have been added and highlighted in yellow in line 52-54.
Point 7: Page 1, line 45 – 46 - The meaning of the sentence is not clear.
Response 7: Thank you very much for your professional opinion, we have modified this and highlighted in yellow in line 55-57.
Point 8: Page 2, lines 59-64 - The purpose of the study must be reformulated in traditional manner.
Response 8: Thank you very much for your professional opinion, we have modified this and highlighted in yellow in line 78-84.
Point 9: Page 2, line 73 - unusual abbreviation for control.
Response 9: Thank you very much for your reminding. “Sub-samples” is equal to the “subsample”.
Point 10: Page 2, Methods – give a detailed description of the determination methods of dry matter, lactic acid, acetic acid, propionic acid, butyric acid, crude protein, true protein and non-protein nitrogen not only reference.
Response 10: Thank you very much for your professional opinion, we have modified it and highlighted in yellow in line 111-115.
Point 11: Page 4, Table 1 – TP and NPN data are not presented in Table 1.
Response 11: Thank you very much for your reminding. We've added to the data of TP and NPN in Table 1.
Point 12: Page 4, lines 167-182 - Remove these sentences from the Results section and place them in the Discussion section.
Point 13: Page 4 - Describe in more detail the results presented in Table 2, not only taking into account the effect of ellagic acid on the studied parameters, but also the dynamics of these parameters during ensiling.
Point 14: Page 4 - Describe in more detail the results presented in Table 3, not only
taking into account the effect of ellagic acid on the studied parameters, but also the dynamics of these parameters during ensiling.
Point 15: Page 4, lines 184-188 - Remove these sentences from the Results section and place them in the Discussion section.
Point 17: Page 9 - In my opinion, it is better to combine the discussion sections into a single text without division into sub-sections with a link to illustrative material.
Response 12-15,17: Thank you very much for your professional opinion. We apologize for the unprofessional writing of the paper and we have revised it. We combine the results and discussion into one part for description in section 3 text.
Point 16: Page 7, Figure 1 - give a transcript of the abbreviation in the figure legend.
Response 16: Thank you very much for your reminding. We have added transcript of the abbreviation in the figure legend.

Reviewer 3 Report
The work is well designed, however, the presentation need improvement.
- Abstract, delete numbers 1,2,3,4 at line 12 and 14 as they are unclear for the reader
- L15-20, the data should be more clear (control and which treatment? Or illustrate the insignificant differences firstly
- In introduction, more data should be clarified about stylo - name, availability, problems ....etc for the respective reader
- Authors should clarify the availability of ellagic acid, cost, and the alternative sources for cheaper process
- L36 Enterobacter ...italic L48, rephrase the sentence stated with 14_16
- L63 add comma after 7
- L84 superscript -1 , -6
- L118, 130. Replace was by was done
- L124,add the respective number for the reference He et al
- L126, Than should be then
- Section 3.2 text at l167-188 should be deleted as these are discussion part and repeated exactly at lines l276-297. The author should add describe the results shown in table 2
- Table 2 please make it more clear to the respective readers, symbols of statistical analysis could be superscript and the data should be presented as the same sequence of the text to be easily followed
- Fig 1 should be enlarged
- L208,209, 210 add , respectively after CK
- L231, add space after d30
- L243 Lactobacillus should be italic
- Figure 5 if possible make italic names for species
- Section 4.4 reconfirmed italic font for all microbial names at L319,321,322,324,326,327,329,332,335,336,338,339,341,343,335,347,348,350,351,352-354,357,359,360,365,366, 371-374
- More in depth discussion about EA effect and expected mechanism should be given
- Conclusion clarify the first sentence about silage
- Confirm all scientific names In reference section for example L385,388,446,451,465,470,479,484,485,488,....etc
Author Response
Response to Reviewer 3 Comments
Thank you very much for your professional opinion. I have carefully read and quoted these suggestions for revision that you sent me. Next, I will answer them separately.
Point 1: Abstract, delete numbers 1,2,3,4 at line 12 and 14 as they are unclear for th
reader.
Response 1: Thank you for your suggestion. We have deleted these numbers and highlighted in red in line 12-14.
Point 2: L15-20, the data should be more clear (control and which treatment? Or illustrate the insignificant differences firstly.
Response 2: Thank you very much for your professional opinion. We have deleted the data that is not clear so that can understand it better and highlighted in red in line 16-18.
Point 3: In introduction, more data should be clarified about stylo - name, availability, problems ....etc for the respective reader.
Response 3: Thank you very much for your reminding. We have added some information about stylo- name, availability, problems ....etc, and highlighted in red in introduction in line 32-38.
Point 4: Authors should clarify favailability of ellagic acid, cost, and the alternative sources for cheaper process.
Response 4: Thank you very much for your reminding. We have added a description of the uses,cost and sources of ellagic acid highlighted in red in line 45-68.
Point 5: L36 Enterobacter ...italic.
Response 5: Thank you very much for your reminding.We have written the format correctly highlighted in red in line 41-42.
Point 6: L48, rephrase the sentence stated with 14_16.
Response 6: Thank you very much for your reminding. We have modified the sentence highlighted in red in line 66.
Point 7: L63 add comma after 7.
Response 7: Thank you very much for your reminding. We have modified this highlightedin red in line 82.
Point 8: L84 superscript -1 , -6.
Response 8: Thank you very much for your reminding. We have modified this highlighted in red in line 104.
Point 9: L124, add the respective number for the reference He et al.
Response 9: Thank you very much for your reminding. We have modified this in line 139 and 143.
Point 10: L126, Than should be then.
Response 10: Thank you very much for your reminding. We have modified this highlighted in red in line 142.
Point 11: Section 3.2 text at L167-188 should be deleted as these are discussion part and repeated exactly at lines L276-297. The author should add describe the results shown in table 2.
Response 11: Thank you very much for your professional opinion. We apologize for the unprofessional writing of the paper and we have revised it. We combine the results and discussion into one part for description in section 3 text.
Point 12: Table 2 please make it more clear to the respective readers, symbols of statistical analysis could be superscript and the data should be presented as the same sequence of the text to be easily followed.
Response 12: Thank you very much for your reminding. We are very sorry for the unprofessional making of the form and have modified it now,in Table 2 and Table 3.
Point 13: Section 4.4 reconfirmed italic font for all microbial names at L319,321,322,324,326,327,329,332,335,336,338,339,341,343,335,347,348,350,351,352-354,357,359,360,365,366, 371-374. More in depth discussion about EA effect and expected mechanism should be given. Conclusion clarify the first sentence about silage. Confirm all scientific names In reference section for example L385,388,446,451,465,470,479,484,485,488,....etc
Response 13: Thank you very much for your reminding. We have modified it highlighted in red.

Round 2
Reviewer 1 Report
I don't have time to review the manuscript.
Author Response
Thanks for your help!